# Matched asymptotic solution for crease nucleation in soft solids

P. Ciarletta [1,2]

A soft solid subjected to a large compression develops sharp self-contacting folds at its free surface, known as creases. Creasing is physically different from structural elastic instabilities, like buckling or wrinkling. Indeed, it is a fully nonlinear material instability, similar to a phase-transformation. This work provides theoretical insights of the physics behind crease nucleation. Creasing is proved to occur after a global bifurcation allowing the co-existence of an outer deformation and an inner solution with localised self-contact at the free surface. The most fundamental result here is the analytic prediction of the nucleation threshold, in excellent agreement with experiments and numerical simulations. A matched asymptotic solution is given within the intermediate region between the two co-existing states. The self-contact acts like the point-wise disturbance in the Oseen's correction for the Stokes flow past a circle. Analytic expressions of the matching solution and its range of validity are also derived.

[1] MOX Laboratory, Dipartimento di Matematica, Politecnico di Milano, piazza Leonardo da Vinci 32, 20133 Milano, Italy. [2] Centre National de la Recherche Scientifique (CNRS), Sorbonne Universités, UMR 7190, F-75005, Paris, France. Correspondence and requests for materials should be addressed to P.C. (email: pasquale.ciarletta@polimi.it)

A soft solid subjected to a large compression can develop sharp self-contacting folds at its free surface, also known as creases. This behaviour is illustrated in Fig. 1 for a nonlinear elastic slab made of a compliant acrylamide gel.

In contrast to the finite-time formation of singular surface cusps in viscous hydrodynamic flows[1], creases in soft solids occur instantaneously when a critical stretch threshold is attained, without any preceding surface furrowing. This singular behaviour displays similar features to phase-transition phenomena. Indeed, it allows the co-existence of two scale-invariant solutions with the remarkable difference of not having an energy barrier from the flat surface to the creased solution[2]. In analogy to liquid–vapour transformations, taking the compressive stretch as the state parameter of the system, creasing in soft solids occurs at a critical binodal threshold occurring before the spinodal limit predicted by Biot[3]. In practice, the free surface is linearly stable to small perturbations up to the Biot threshold but it can become metastable at lower compression, thus bifurcating into a localised singular solution that is able to release the overall elastic free energy.

Creasing has attracted a lot of interest not only from engineering sciences, especially for the possibility to fabricate devices with adaptive surface morphology at different length-scales[4–6], but also from developmental biology, in order to explain the formation of localised furrows during either tissue morphogenesis[7], e.g., the convolutions of the brain[8], or tumour development[9]. The problem of detecting the critical stretch for crease nucleation has been the focus of many recent numerical studies, in which the main challenge was to reproduce a spontaneous breaking of the scale symmetry in finite element simulations. This has been done either by introducing a stabilising bending energy at the free surface, and then looking for the limiting behaviour whilst letting this additional term goes to zero[10], or by enforcing self-contact in a small portion of the free surface, and then looking for the energy difference between the creased and the smooth solution[11]. Numerical estimates for the critical nucleation threshold are reported, in good agreement with the experimental data[12].

Unlike similar material instabilities in solids, such as cracking[13], twinning[14] and cavitation[15], the nucleation condition for creasing is still unexplained theoretically. Difficulties arise since the free elastic energy of a nonlinear elastic solid contains derivative nonlinearities. In such a particular case, the Weierstrass criterion of positiveness of the second variation of the energy functional is not sufficient for material stability. Thus, even when considering a solid with a smooth, strongly elliptic elastic energy, singular solutions can nucleate as localised creases[16]. Such discontinuous solutions represent strong marginally stable states, that appear locally and are independent on the geometry. They can occur not only as a consequence of the loss of energy convexity introduced by the incompressibility constraint, but also in compressible solids[17]. A more restrictive necessary condition, quasi-convexity[18], is related to the existence of minimal solutions nucleating in the inner part of the domain. Nonetheless, only in very special cases does this result of calculus of variations provide an operative rule to build a general criterion of global stability, e.g., in solid–solid phase transformations[19]. Moreover, creasing is a problem of additional complexity, since quasi-convexity must be imposed at the Neumann part of the boundary[20,21].

In summary, although much progress has been done in the last years in the comprehension of the creasing behaviour, a complete understanding of the physics behind crease nucleation remains elusive. This works derives an analytic criterion of crease nucleation by making use of bifurcation techniques and singular perturbation theory. After defining the elastic problem, the crease nucleation threshold is calculated analytically from the necessary and sufficient condition for the onset of a global instability. A matched asymptotic approximation of the crease solution is later derived and validated against experimental and numerical results.

## Results

**The nonlinear elastic model.** We consider an elastic half-plane subjected to a planar deformation with a controlled stretch in the horizontal direction. Dealing with an intrinsically scale-free instability, it is indeed expected that the results will also be relevant to the finite-size problem, because the crease scaling makes every free surface behave locally as a half-plane. The vectors $\mathbf{X} = \mathbf{X}(R, \Theta)$ and $\mathbf{x} = \mathbf{x}(r, \theta)$ denote material and spatial positions in cylindrical polar coordinates, respectively. The coordinate system is centred at the crease nucleation point, and the free surface in the material configuration is described by $\Theta = \pm \pi/2$, as illustrated in Fig. 2.

The kinematics is described by the geometrical deformation tensor $\mathbf{F} = \partial \mathbf{x}/\partial \mathbf{X}$.

The elastic body behaves as an incompressible Neo-Hookean solid, so that its strain energy density reads:

$$W = \frac{\mu}{2}(\mathrm{tr}(\mathbf{F}\mathbf{F}^T) - 2) - p(\det \mathbf{F} - 1); \qquad (1)$$

where $\mu$ is the shear modulus, and $p$ is the Lagrange multiplier enforcing the incompressibility constraint. Neglecting the body forces, the elastic equilibrium equation reads:

$$\mathrm{div}\ \boldsymbol{\sigma} = 0; \qquad (2)$$

where $\boldsymbol{\sigma} = \mu \mathbf{F}\mathbf{F}^T - p\mathbf{I}$ is the Cauchy stress tensor, $\mathbf{I}$ being the identity matrix. The nonlinear elastic boundary value problem must be complemented by different boundary conditions at the free surface for each of the two homogeneous states that are assumed to co-exist after crease nucleation, as hinted in[2].

Firstly, the material is subjected to a uniform stretch $\lambda_x$ along the horizontal direction far away from the crease, where the top surface is free of external traction. In this case, the boundary value problem is made by Eq. (2) with the following Neumann conditions at the free surface:

$$\sigma_{r\theta} = \sigma_{\theta\theta} = 0, \quad \Theta = \pm \pi/2; \qquad (3)$$

and it is solved by an affine mapping given by the following outer

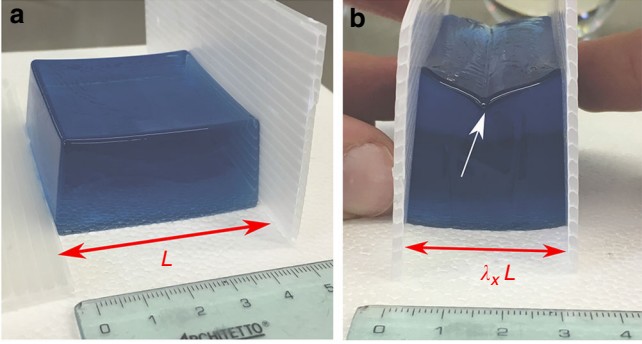

**Fig. 1** Crease formation in a compressed soft slab. **a** A $5 \times 5 \times 2\ \mathrm{cm}^3$ soft block is made by mixing 37.2 mL of deionised water containing Blue Dextran (1 mg/mL) with 16 mL of 0.5 M Tris–HCl buffer at pH 6.8, and 8.75 mL of a solution containing Acrylamide (30% w/v) and Bis-acrylamide (0.8% w/v) in a soft silicone-based mould. In total 625 μL of APS solution (10% w/v) and 125 μL TEMED were gently mixed to the gel-forming mixture, which was allowed to polymerise for 1 h. **b** The gel is then gently pulled out from the mould and axially compressed by hand using polystyrene sheets to avoid friction. The white arrow indicates the finite crease at the free surface observed at an axial stretch of about $\lambda_x \simeq 0.6$

solution:

$$r_{\text{out}} = \left(\lambda_x^2 \sin^2\Theta + \lambda_x^{-2}\cos^2\Theta\right)^{1/2} R \tag{4}$$

$$\theta_{\text{out}} = \tan^{-1}\left(\lambda_x^2 \tan\Theta\right); \quad p_{\text{out}} = \mu\lambda_x^{-2};$$

where the subscript indicates that the fields pertain to the outer domain. In particular the top surface remains flat, being $\theta = \pm\pi/2$, and the controlled stretch is sustained by a uniform horizontal stress $\sigma^{\text{out}}$ given by:

$$\sigma^{\text{out}} = \sigma_{rr}(r, \pm\pi/2) = \mu\left(\lambda_x^2 - \lambda_x^{-2}\right). \tag{5}$$

Second, the inner creased solution allows a frictionless self-contact at the top surface, so that:

$$\sigma_{r\theta}(r, \pm\pi) = 0; \quad \sigma_{\theta\theta}(r, -\pi) = \sigma_{\theta\theta}(r, +\pi) \quad r \leq r_c. \tag{6}$$

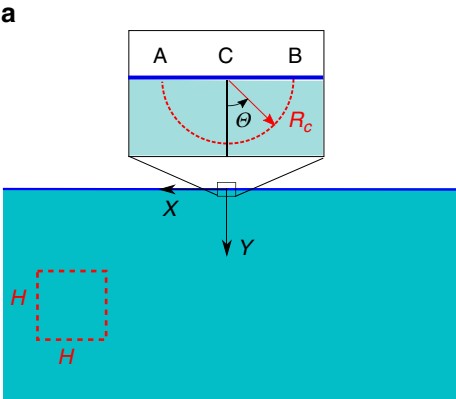

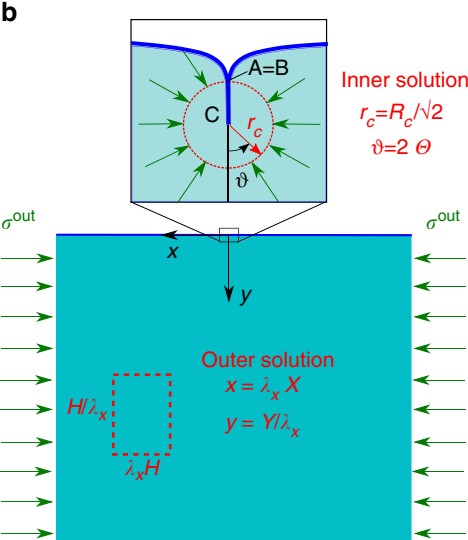

**Fig. 2** Definition of the elastic model. Sketch of the reference (**a**) and current (**b**) configurations, with the corresponding coordinate systems, of a soft elastic half-space subjected to a controlled finite axial stretch $\lambda_x$, sustained through an axial compressive stress $\sigma^{\text{out}}$. The insets show the inner solution, defining a small crease nucleating in the point C, in which an half circle of radius $R_c$ is mapped into a full circle of radius $r_c$ (dotted red lines), with self-contact along the lines AC and AB. The far-field outer solution defines an affine transformation, deforming a square of length $H$ into a rectangle of sides $\lambda_x H$ and $H/\lambda_x$ (dashed red lines)

The continuity of the hoop stress across the self-contact line is dictated by assuming a mirror symmetry across the vertical axis passing through the nucleation point and by neglecting a surface energy contribution. Being subjected to a unilateral constraint, it is also assumed that the hoop stress $\sigma_{\theta\theta}$ in Eq. (6) is everywhere non-positive within the creased domain.

As remarked in[16], the boundary value problem given by Eqs. (2, 6) admits a solution that is invariant by re-scaling, i.e., $\mathbf{x}(\mathbf{X}) = R_c\,\mathbf{x}(\mathbf{X}/R_c)$, and by horizontal translation along the free surface. Such a mapping transforms a half-circle of radius $R_c$ in the material configuration into a full circle with radius $r_c = R_c/\sqrt{2}$ with self-contact along a line of length $r_c$, as shown by the dotted lines in the insets of Fig. 2.

This inner solution is given by:

$$r = R/\sqrt{2}; \quad \theta = 2\Theta; \quad p = \frac{\mu}{2} - \sigma_{rr}(r_c) - \frac{3}{2}\mu\log\left(\frac{r}{r_c}\right); \tag{7}$$

where $\sigma_{rr}(r_c)$ represents the hydrostatic pressure that the outer solution exerts on the crease. Eq. (7) belongs to the class of universal solutions for isotropic nonlinear elastic materials[22,23], resulting in a logarithmic stress singularity at the nucleation point.

The creased half-plane is characterised by a crossover between the co-existing scale-invariant solutions given by Eqs. (4, 7), since such homogeneous states are geometrically incompatible. In contrast to previous numerical works, singular perturbation techniques will be used in the following to match analytically these inner and outer solutions.

**Crease nucleation threshold**. The above inner and outer solutions are intrinsically incompatible from a geometric viewpoint. A half-circle in the material configuration is indeed mapped into a full circle by the former and into a half-ellipse by the latter. Since they cannot be matched directly, an inhomogeneous solution is expected to emerge in an intermediate domain when the compressive stretch reaches the threshold $\lambda^{\text{cr}}$ for nucleation. In the following, the creasing threshold is sought by investigating the necessary and sufficient conditions for the existence of such a matching solution.

Trivial estimates of the upper and lower bounds for the nucleation threshold can be derived by simple physical considerations.

A trivial lower bound $\lambda_x^{\text{lb}}$ is given by the loss of local stability with respect to weak variations, i.e., perturbations that are small together with their derivatives. This problem has been solved by the Biot, giving the limit $\lambda_x^{\text{lb}} \simeq 0.544$. In the post-buckling regime, the nonlinear resonance between all modes seems to drive the formation of crease-like static shocks beyond this critical value[24,25]. Notwithstanding, experiments have reported that creases nucleate much before this limiting stretch is attained. This means that at a higher stretch the body becomes unstable with respect to local strong variations, i.e., local perturbations with small amplitudes but non-small derivatives.

A trivial upper bound $\lambda_x^{\text{ub}}$ can be estimated by recalling that the creased solution must globally have a lower strain energy compared to the homogeneous outer compression. Since all weak perturbations of the outer solution locally increase its strain energy density for $\lambda > \lambda_x^{\text{ub}}$, it is necessary that the inner solution locally possess a lower elastic energy. Taking into account the principal stretches of both inner and outer solutions, this condition requires that $\lambda_x^{\text{ub}} = 1/\sqrt{2} \simeq 0.707$.

Since the body is linearly stable to weak perturbations in the range $0.544 < \lambda_x < 0.707$, a necessary condition for creasing is that the body becomes metastable with respect to a localised rotation at the free surface, taken large enough to create a self-contact.

Thus, creasing occurs after a global bifurcation in the configurational space, being mathematically associated with the loss of strong local minimum.

In nonlinear ordinary differential equations, the onset of a inhomogeneous solution connecting two fixed points can be proved by showing the existence of a heteroclinic trajectory between them[26]. The fixed points represent two homogeneous states of the original problem, none of which agrees with all side conditions. The same underlying principle can be adapted to the elastic boundary value problem for establishing an operative rule to detect crease nucleation. The scale-free inner and outer states can not be matched directly. Thus, a criterion for crease nucleation can be sought by proving the existence of a inhomogeneous displacement field matching the perturbative series of the inner and outer solutions. A necessary condition for creasing is given by the loss of marginal stability of the inner solution. Indeed, it opens the possibility for the existence of an adjacent inner solution that can be matched with the far–away displacement and pressure fields of the perturbed outer solution.

This necessary condition is studied using the theory of incremental elastic deformations superposed on a finite strain[27]. Let the inner solution be confined in the spatial domain given by $0 \leq r \leq r_c$, in which the crease radius $r_c$ is considered small compared to any characteristic length in a finite-size problem. The basic inner solution is given by Eq. (7). Recalling the axial symmetry of the inner solution, the horizontal force exchanged across the vertical line bisecting the crease is given by $2\int_0^{r_c} \sigma_{\theta\theta}dr = 2r_c\,\sigma_{rr}(r_c)$). Enforcing the balance of forces between the inner solution and the outer uniform traction exerted at the side wall gives $\sigma_{rr}(r_c, \theta) = \sigma^{out}$. It is important to highlight that this boundary condition corresponds to the application of a dead load, since this applied traction only exists in the spatial configuration, not having any material counterpart. Let $\delta\mathbf{x} = [u(r,\theta), v(r,\theta)]^T$ be an incremental deformation vector superposed on the inner solution, i.e., $|\delta\mathbf{x}| \ll r_c$, with radial and tangential fields $u$ and $v$, respectively. Let $\mathbf{\Gamma} = \text{grad}(\delta\mathbf{x})$, where the gradient operator is referred to the finitely deformed inner solution. The incompressibility constraint imposes:

$$\text{tr } \mathbf{\Gamma} = 0. \tag{8}$$

By standard Taylor expansions around the inner solution, the components of the push-forward $\delta\mathbf{S}$ of the incremental Piola-Kirchhoff stress tensor read:

$$\delta S_{ji} = A_{jikl}\Gamma_{lk} + p_{in}\Gamma_{ji} - \delta p_{in}\delta_{ji}; \tag{9}$$

with $(i, j, k, l) = (r, \theta)$, $\delta p_{in}$ being the increment of the Lagrange multiplier, $\delta_{ji}$ the Kronecker delta and $A_{jikl}$ the elastic instantaneous moduli. The incremental equilibrium equations are given by:

$$\text{div } \delta\mathbf{S} = 0; \tag{10}$$

with the following boundary conditions at the free surface:

$$\delta S_{rr} = \delta\sigma_{rr} = \frac{3\mu u}{2r_c}; \qquad \delta S_{r\theta} = 0 \quad \text{at } r = r_c; \tag{11}$$

where $\delta\sigma_{rr}$ is the increment of the external traction caused by the perturbation of the interface. The incremental boundary value problem given by Eqs. (8, 10, 11) can be solved by enforcing the boundedness of the incremental displacement and stress fields at the crease nucleation point $r = 0$ for the sake of physical compatibility. Considering the periodicity imposed by self-contacting, the solution is sought using normal modes such that $u(r,\theta) = U(r)\cos(m\theta)$, $v(r,\theta) = V(r)\sin(m\theta)$, where $m$ is the integer circumferential wavenumber. The dispersion relation giving the marginally stable mode $m$ as a function of the order parameter $\lambda_x$ is found after standard manipulations of the incremental equations (see Supplementary Note 1). Most interestingly, the inner solution loses its local uniqueness for $m = 2$ at the following value of critical outer stress:

$$\begin{aligned}\sigma^{out} &= \frac{\mu}{8}\left(1 - 3\sqrt{5} + \sqrt{18 - 2\sqrt{5}}\right. \\ &\left.- 2\sqrt{26\sqrt{5} - \sqrt{62\sqrt{5} + 1038} + 28}\right) \simeq -2.053\mu.\end{aligned} \tag{12}$$

From Eq. (5), the necessary condition in Eq. (12) corresponds to a precise surd prediction of the crease nucleation threshold $\lambda^{cr}$, such that:

$$\lambda^{cr} \simeq 0.637554 \tag{13}$$

Eq. (13) is the first theoretical insight giving an analytic criterion about the critical stretch for crease nucleation. This predicted threshold is in excellent agreement with both experimental and numerical investigations, reporting a value in the range $0.63 - 0.643$[2,12]. In particular, the theoretical prediction is strikingly close to the experimental value of 0.635, that has been recently measured with a high resolution imaging technique[28].

In order to prove that this necessary condition is also sufficient for creasing, so that Eq. (13) is an actual prediction of the nucleation threshold, matched asymptotic theory will be used to show the existence of a inhomogenous solution matching the perturbative series representing the inner and outer states.

**Near-field solution in the intermediate region**. The geometrical incompatibility between the outer and inner solutions has a striking analogy with the 2D Stokes paradox of the uniform viscous flow around a circle[29]. The similarities are discussed in detail in the following.

Let $H$ be a macroscopic characteristic length for a finite-size block. The outer solution is displayed at a very large distance $r \gg r_c$ from the nucleation point, where the effect of the presence of a small crease becomes negligible. Thus, the outer solution of this elastic problem plays the role of the uniform flow solution in the hydrodynamic problem very far away from the obstacle. As for the Oseen's correction[30], let us investigate the existence of a near-field matching solution at a typical length-scale $\tilde{r}$, such that $r_c \ll \tilde{r} \ll H$.

After crease nucleation, the elastic boundary problem changes radically, since the a localised portion of the free surface is no longer free of tractions. Assuming a frictionless contact, there is an exchange of a distributed normal traction across the self-contact line, with a zero net force in the current configuration. However, by pulling back this self-contact traction to the homogeneously deformed state, a vertical distributed pressure is obtained on a flat portion of the free surface in the finitely deformed configuration. If a near-field solution is sought sufficiently far away from the crease, the distributed pressure can be approximated by a vertical concentrated force $F$ at the crease nucleation point, given by:

$$F = 2\int_0^{r_c}\sigma_{\theta\theta}\,dr = 2\,r_c\,\sigma^{out}; \tag{14}$$

whose magnitude scales as the crease radius. Thus, the perturbative outer series takes into account for the vertical compression due to the localised surface folding superposed over the homogeneous state of horizontal compression. In practice, the

self-contact in this elastic problem has an analogous effect to the flow disturbance of the circle in the Oseen problem. Since for an incipient crease $F = O(r_c)$ has a small magnitude compared to the total axial force necessary to sustain the uniform stretch on the side walls, the elastic solution in the intermediate region can be found by superposing the singular force $F(x, y) = F\,\delta(x, y)$, where $\delta(x, y)$ is the delta function, over the affine outer solution given by Eq. (4). Transforming to Cartesian coordinates as in Fig. 2, an elastic stream function $\Psi$ is introduced to enforce incompressibility, such that the incremental displacement vector components $u_y$ and $u_x$ are given by:

$$u_x = -\Psi_{,y}; \qquad u_y = \Psi_{,x}. \qquad (15)$$

After standard manipulations[31], the incremental equilibrium condition in Eq. (10) reads:

$$\nabla\overline{\nabla}\Psi = 0; \qquad (16)$$

where $\nabla = (\partial_{xx} + \partial_{yy})$ and $\overline{\nabla} = (\partial_{XX} + \partial_{YY})$ are the Laplacian operators in the stretched $(x, y)$ and reference $(X, Y) = (x/\lambda_x, y\,\lambda_x)$ coordinates, respectively. Since we deal with a point force at the origin, the stream function is given by the Green's function solving Eq. (16) for the half-plane[32]. Imposing a zero shear stress component at the free surface $y = 0$, the solution for the incremental displacement fields reads:

$$u_x = -2a_0\left(\left(\lambda_x^4 + 1\right)\tan^{-1}\left(\frac{x}{y}\right) - 2\lambda_x^2\tan^{-1}\left(\frac{x}{\lambda_x^2 y}\right)\right); \qquad (17)$$

$$u_y = a_0\left(\left(\lambda_x^4 + 1\right)\log\left(\frac{x^2 + y^2}{\overline{H}^2}\right) - 2\,\log\left(\frac{\lambda_x^4 y^2 + x^2}{\overline{H}^2\lambda_x^2}\right)\right). \qquad (18)$$

The characteristic length $\overline{H}$ in Eq. (18) re-scales the maximum vertical displacement $h$ at the free surface, i.e., the crease depth. The assumption that $h = O(r_c)$ for an incipient crease gives an order of magnitude for $\overline{H}$, but fixing its value is beyond a linear stability analysis. The parameter $a_0$ can instead be calculated by imposing overall vertical equilibrium under the action of the concentrated load[33], as:

$$a_0 = \frac{r_c\left(\lambda_x^2 + 1\right)}{\pi\left(\lambda_x^6 + \lambda_x^4 + 3\lambda_x^2 - 1\right)}. \qquad (19)$$

In particular, it can be observed that $a_0$ diverges when the denominator at the rhs of Eq. (19) vanishes, which happens exactly at the Biot threshold, since all surface undulations become unstable even in the absence of a singular force. From a mathematical standpoint, this situation implies an incompatibility between the principal parts of the linearised elliptic differential operator and the corresponding boundary operator at the free surface, i.e., a violation of the complementing condition[34].

At a distance $r \gg r_c$ from the crease, the incremental displacement and pressure fields become negligible, thus automatically matching the outer solution. In proximity of the inner solution, i.e., for $\overline{r} \to r_c$, the near-field displacement and pressure fields become of order $O(1)$, the perturbative expansion becomes ill-posed and a direct matching cannot be achieved. However, the near-field approximation shows that displacement and pressure fields of the inner and outer solutions become geometrically and physically compatible in proximity of the crease boundary (see Supplementary Note 2).

The range of validity for the intermediate solution is finally investigated. A lower bound is given by enforcing that the incremental displacements be much smaller than the ones corresponding to the outer solution, obtaining that $\overline{r} \gg r_c$. The upper bound results from the global area-preservation of the deformed body in the intermediate region. Indeed, since the lower bound excludes the singularity of the logarithm at the nucleation point in Eq. (18), it can be shown that the global area is preserved if and only if $\overline{r}/\overline{H} \ll r_c/\overline{H}\left|\log(r_c/\overline{H})\right|$. In summary, the asymptotic matching with the outer and inner solutions imposes the following restrictions for the dimensionless intermediate variable $\frac{\overline{r}}{\overline{H}}$:

$$\frac{r_c}{\overline{H}} \ll \frac{\overline{r}}{\overline{H}} \ll \frac{r_c}{\overline{H}}\left|\log\left(\frac{r_c}{\overline{H}}\right)\right| \ll 1; \qquad (20)$$

that define the domain of validity of the near-field solutions given by Eqs.(17,18). It is worth noticing that the description of the crease opening at the free surface is out of reach of this model, since it occurs in a domain situated outside the ranges of validity of both the inner and the intermediate solution.

**Mixed finite element simulations**. The boundary value problem is then solved numerically using a mixed finite element formulation. The simulations are performed using the software ABAQUS 6.14.1 (Dassault Systems Simulia Corp., Providence, Rhode Island, USA). A block of finite thickness $L$ is discretized imposing a homogeneous horizontal stretch $\lambda_x$ on the side walls, preventing the vertical displacement of the bottom line. The horizontal width is taken much bigger than the thickness in the reference configuration, in order to neglect the boundary effects of the lateral block walls on the fully developed creased regime. The block is discretized using a structured mesh made of 17,600 hybrid, four-node bilinear elements, providing a suitable stability range in mixed formulations of finite elasticity[35]. Self-contact is allowed at the free surface, yet not prescribed a priori, with a tangential frictionless contact based on a finite sliding algorithm not allowing overdisclosure. A nodal compressive force of small amplitude is incrementally imposed in the middle of the free surface, simultaneously with an horizontal stretch on the block sides, in order to obtain a single crease in a weakly nonlinear regime at $\lambda_x = 0.62$. The nodal force is then removed whilst keeping the stretch constant. Thus, the crease nucleation and its morphology are studied by gradually and completely removing the axial compression whilst keeping the nodal force equal to zero. An adaptive, iterative increment algorithm is set for both the nodal force and the applied strain. Since creasing is a subcritical instability, a pseudo-dynamic method is introduced[36]. An automatic stabilisation scheme with a small damping factor $\gamma$ is implemented in the direct solver using a Full Newton technique.

The resulting numerical displacement fields are compared to the theoretical predictions of the inner, outer and intermediate solutions in Fig. 3. In particular, the panel e in Fig. 3 illustrates that the vertical displacement at the free surface predicted by Eq. (18) perfectly fits the numerical result without any adjustable parameter, within the whole predicted range of asymptotic validity given by Eq. (20).

The detection of the crease nucleation point has been performed by progressively reducing the damping coefficient down to the minimal value at which the simulated creased slab is able to jump back to the homogeneous outer solution whilst reducing the axial compression beyond the critical point (see Supplementary Note 3). Figure 4 finally displays the ratio of the simulated total strain energy of the block $E_{\text{num}}$ over the corresponding value $E_{\text{hom}}$ for the basic homogeneous solution at different damping values, so that the crease nucleation point is

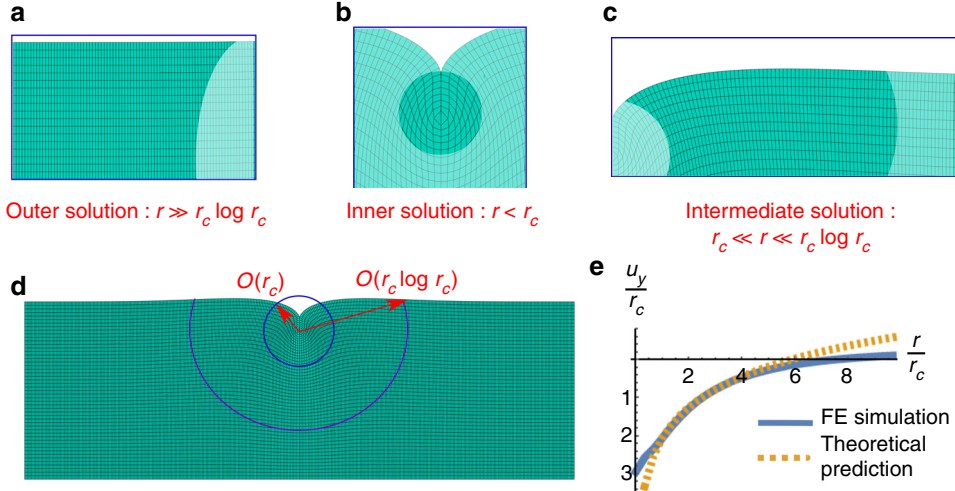

**Fig. 3** Finite element simulations. Results of the mixed finite element simulation of an elastic slab of initial unit thickness displaying a crease with radius $r_c = 0.04$ and depth $h = 2r_c$ at an axial compression just beyond the nucleation threshold, i.e., $\lambda_x = 0.63$. The outer, inner and intermediate solutions are depicted in **a–c**, highlighting the respective range of validity in the asymptotic analysis. The overall deformed shape of the elastic slab is shown in **d**, with indication of the underlying mesh. In **e**, the numerical near-field vertical displacement (blue solid line) is fitted by the theoretical prediction of Eq. (18) (dashed orange line) without any adjustable parameter

given by the critical stretch at which this ratio becomes smaller than one. In particular, by decreasing the damping factor down to its minimal value allowing numerical convergence, the value of nucleation threshold is numerically found at $\lambda_x^{cr} = 0.6372$, in excellent agreement with the analytic prediction of Eq. (13). In summary, the numerical results fully validate the theoretical predictions about the crease nucleation threshold and the matched asymptotic solution.

## Discussion

Creasing is a material instability, meaning that it nucleates locally in space and does not depend on the global geometry of the body. This work demonstrates indeed that crease nucleation results from a global instability in the configurational space, allowing the co-existence of an affine outer deformation and an inner solution with localised self-contact at the free surface.

An analytic criterion for nucleation is formulated by showing that a creased solution can exist if and only if the inner state loses its marginal stability, so that the inner and outer solution can be matched. Accordingly, Eq. (13) gives a theoretical prediction of the critical stretch for crease nucleation, in excellent agreement with previous numerical and experimental studies. The loss of marginal stability of the inner solution is a fundamental understanding on the physics behind the onset of creasing. Creasing is ubiquitous since the inner deformation is a universal solution in finite elasticity, meaning that it can be supported in equilibrium for every isotropic material by suitable surface tractions alone. Thus, the creasing threshold specific to a different constitutive equation can be calculated by solving the corresponding inner incremental problem, exactly as it is shown in this work. This operative rule can also be useful in other system models where a nucleus spontaneously appears at a free boundary.

The existence of a inhomogeneous creased solution is shown using a matched asymptotic approximation, constructing analytically a near-field solution in the intermediate region between the co-existing inner and outer homogeneous states. The outer perturbative problem becomes singular because of the surface self-contact in the inner creased domain, acting like the point-wise disturbance in the Oseen's correction for the 2D Stokes problem of the flow past a circle. Using Green's functions in the half-space,

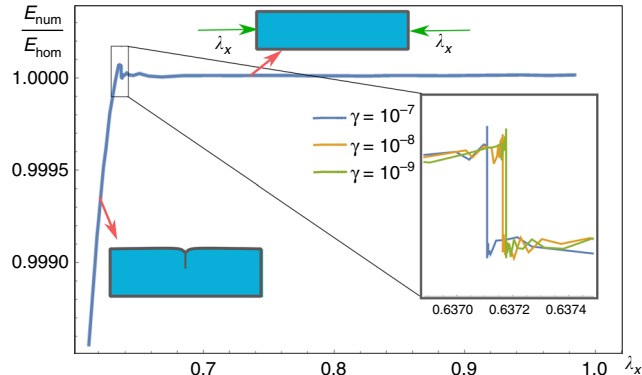

**Fig. 4** Numerical results. The solid curves depict the ratio of the total elastic energy $E_{num}$ of the numerical simulation and the corresponding energy $E_{hom}$ of the homogeneously compressed slab at different values of the damping coefficient $\gamma$. The inset shows that the resulting nucleation limit tends to the critical threshold of $\lambda^{cr} = 0.6372$ whilst decreasing the damping factor, in excellent agreement with the prediction of Eq. (13)

a matching solution is given by Eqs.(17,18) within a range of validity of the intermediate variable given by Eq. (20). This analytic prediction perfectly fits the results of numerical simulations without any adjusting parameter.

This scale-invariant approximation is unable to resolve the indeterminacy on the crease radius and depth, since an infinite degeneracy of self-similar creases becomes available due to the lack of an energy well for crease nucleation. Adding a regularising effects, such as a surface energy[37], would introduce an energy barrier that automatically selects the creased state with minimal energy. For this purpose, the proposed perturbation analysis should be pushed to higher orders of approximation, since nonlinearities become dominant in pattern formation[16]. Moreover, the proposed perturbation method is unfit for deriving the scaling law that governs the cusped profile of the free surface in vicinity of the crease opening. Another open challenge is unravelling the interaction mechanisms between multiple creases resulting from heterogeneous nucleation, due to the high

sensitivity of creasing to imperfections. In fact, multiple creases can nucleate at distant sites to minimise the total elastic energy in a finite–size body, as observed experimentally in soft slabs of different geometry under compression[28].

In conclusion, this work fosters our physical understanding of creasing, presenting important insights for pushing key theoretical advances enabling technological progress. For example, the steps towards the comprehension on how to control pattern formation and channelling on-demand represent a potential breakthrough in the design and the fabrication of the next generation of morphable meta-materials.

**Data Availability**. The author declares that all data supporting the findings of this study are available within the article and its Supplementary Information, or are available from the author upon request.

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

## Acknowledgements

The experiment in Fig. 1 was performed by Wanda Celentano and Francesco Cellesi (Department of Chemistry, Politecnico di Milano). This work was supported by the AIRC MFAG grant 17412.

## Author contributions

P.C. conceived the study, performed the theoretical and the numerical analysis, and wrote the manuscript.

## Additional information

**Competing interests:** The author declares no competing financial interests.

