## [Peer Review File · Nature Communications]

Reviewers' comments:

Reviewer #1 (Remarks to the Author):

This paper address the theory of creasing in highly compressed elastic solids. Creasing occurs when a soft solid body with a free surface is compressed. At a critical degree of compression, the free surface transitions from being flat to having "sharp self-contacting folds ... known as creases". Creasing is ubiquitous in the mechanics of soft solids, and of considerable importance. However, although creasing has been well understood numerically via finite element calculations, there is no analytic theory of creasing. This is surprising, since it appears to be part of the family of elastic instabilities, and there is a standard theoretical toolbox for understanding elastic instabilities based on perturbative linear stability analysis, but these tools fail to capture creasing. The difficulty is easy to identify: a centre of a crease is a cusp with divergent strain/stress, which is not a small perturbation from the flat state of the free surface, even if the amplitude/displacement of the crease is very small. Creasing is thus a new type of "non-linear instability", and finding an analytic description of creasing is certainly one of the most important open questions in the theory of mechanical instabilities, both because creasing itself is important, and because it appears to be a new category of instability.

Here the author claims to have found an analytic description of crease nucleation. The centerpiece of this theory is a prediction of the degree of compression required to nucleate a crease which, for the simplest (neo-Hookean) constitutive law the author is able to produce a surd (14) which amounts to compression by 36.2..%. This prediction is convincingly close to the values that previous studies have found numerically. However, although I can follow the calculation presented by the author to produce this value, I don't understand how the calculation connects to the nucleation of creases.

To be more precise, the author imagines the center/cusp of a crease as following a well known solution in elasticity that corresponds to taking a semi-circle of (2-D) elastic material, and bringing together the two radii to form the semi-circle into a circle. Similarly, far from the crease, the author asserts the elastic body has a homogeneous compression, which is the "outer" solution. For small degrees of compression, the entire body is homogeneously compressed, but when the threshold for creasing is reached, a crease nucleates consisting of an central region of the "inner solution" which somehow matches onto the outer homogeneous solution. This picture was introduced in the second paper by Hohlfeld and Mahadevan, and is likely correct.

To derive the threshold for cusping, the standard approach would be to consider the linear stability of the homogeneous solution, since this is the solution occupied by the elastic body prior to creasing, which becomes unstable at the creasing threshold. This was done by Biot in the 1960s, who showed that the homogeneous solution becomes catastrophically unstable at about 45% compression, however the creasing occurs earlier, at around 35% compression, when the homogeneous solution is certainly linearly stable. Here the author instead considers the stability of the "inner" solution. He considers a semicircle of material wrapped into a circle of radius r_c , following the well-known inner solution, with the point at r_c being the point where the crease ceases self contact and starts to open up. This solution has an unknown overall offset in its pressure field, which the author determines by demanding that the stress at the point r_c match the macroscopic compressive stress in the homogeneous solution. The author then adds a small displacement perturbation that looks like $\cos(m\theta)$, where $0 < \theta < 2\pi$ is the angular coordinate in the circular state. The author linearizes the equations of elastic equilibrium within the inner solution, and finds the form of small perturbation that solves the equations to linear order. He then considers the "boundary" of the inner solution as a circle, and applies the boundary condition that the stress on the boundary after perturbation should be the same as the stress in the unperturbed inner-solution state but at the perturbed location of the boundary.

The author solves the above stability problem, producing the curve Fig 1 in the SI which maps the compressive stress in the homogeneous solution required to make the inner solution unstable for each value of m . I have reproduced this calculation, and it is correct, though I believe there are a few serious typos in the SI

SI eqn 9/10 -> this equation is a linearization in U , but has a $4UU'$ term in?

SI eqn 11 -> I think this is wrong by an overall minus sign?

SI eqn 12 -> I think there are several errors here, and consequently the surd in (13) doesn't solve (12) for $m=2$? I think 12 should be " $m(20(-a+b)+29(-a+b)m^2+(-6+7a+7b)m\sqrt{40+9m^2})-4(10(a-b)m+7(-a+b)m^3-(1+a)(1+b)\sqrt{40+9m^2}+(1+a+b)m^2\sqrt{40+9m^2})\text{sout}-8(a-b)m(-1+m^2)\text{sout}^2=0$ ", which does indeed give the authors surd as a root for $m=2$.

Despite these typos, the calculation does seem to be correct. However, fundamentally I don't understand what the solution to this stability problem has to do with the creasing instability. In particular, I would ask the author to address the following questions:

1) Why does the stability of the inner solution relate to the nucleation of creases at all? The inner solution is only seen after the crease has formed, so what does its stability have to do with anything? Can you make a clear claim that the threshold derived here is an upper bound, lower bound or actual calculation of the threshold for the crease forming instability? The \sim sign in (13) leaves me unsure how precise a claim is being made. I also don't understand the analogy to a heteroclinic trajectory in a dynamical system - what is the actual argument for the stability of the inner solution being relevant?

2) Where do the boundary conditions on the inner solution come from? Why are they the appropriate boundary conditions for the stability analysis, rather than, say, that the boundary stress does not change upon perturbation, or that the stress match the intermediate solution? Similarly, why is a circular (/semicircular) boundary the appropriate boundary to consider?

3) In the manuscript, you write "The basic inner solution is given by (7), whilst the axial symmetry and global equilibrium impose $\sigma_{rr}(r_c, \theta) = \sigma_{out}$." I can't see why global equilibrium imposes this - can you elaborate? Indeed, I would have thought $\sigma_{\theta\theta}(r_c, \pi) = 0$ would be more appropriate, since this is the point the self contact ends and the surface becomes a free surface?

4) Why do you not consider an $m=1$ mode, which becomes unstable at $\sigma_{out} = -\sqrt{7}/2 = -1.32\dots$ (i.e. before the $m=2$ mode) or indeed an $m=1/2$ mode which might have the effect of opening out the inner solution into a crease? I understand that $m=1$ breaks reflection symmetry, but surely the symmetry should emerge from the calculation not be fed into it?

In the latter part of the manuscript, the author discusses matching the inner solution to the outer solution by finding an intermediate solution to the equations of elasticity, in a manner reminiscent of the resolution to the Stokes paradox. Although this is surely a good idea, I'm skeptical about the implementation here. Firstly, the intermediate solution is taken as the Green's function for a point force applied at the middle of the crease. The stress fields in this Green's function transmit an actual, non-zero, vertical force through the elastic body, as can be seen by integrating the stress around the surface of any portion of the body including the force, yet there is no actual vertical force applied at the heart of a crease, so any attempt to match this intermediate solution to the inner solution will end up having stress divergence (i.e. force) at the interface between the solutions - I would have thought a force dipole would be a better matching solution? The author doesn't run into this problem because, as far as I can see, he never actually matches the inner and intermediate solutions since, as seen in (20), the range of validity of the intermediate solution ($r > r_c$) does not overlap with the range of

validity of the inner solution ($r \sim r_c$).

Overall, I have very mixed thoughts about this paper. On one hand, the calculation clearly draws together relevant components of the problem, and the match between the predicted and numerically measured threshold is convincing. On the other hand, I simply don't understand how the author's stability analysis relates to the problem of crease nucleation. Beyond this general lack of understanding, I'm particularly confused about how/why the boundary conditions are chosen, how the inner solution pressure constant is chosen, and how the intermediate solution can ever match the inner or outer solution given it implies a body force within the crease which isn't actually there.

This is a sufficiently important problem that, despite my doubts about the correctness of this work, I am very sympathetic to the idea it should be published for the community to reflect on, and Nature Communications is a reasonable journal for it to be published in. However, in my view the author needs to clarify the above issues and make a clear physical argument for why this calculation relates to crease nucleation. Alternatively, the author could simply address the more technical questions (2-3-4 above, and the question about the body force implied by the intermediate solution) and then add some prominent caveats to the manuscript, walking back a little from the claim that "This work solves the quest for a theoretical explanation of crease nucleation" and present it as a potentially important step towards a theoretical solution, but one that remains fundamentally mysterious.

Reviewer #2 (Remarks to the Author):

This article reports analytical study of the formation of creases at the surface of an elastic soft solid under compression. The article also includes comparison with the results with a numerical simulation.

The author provides here an analytical prediction for the threshold of creasing.

This is interesting because the system was shown to be linearly stable up to a large compression (Biot, 1963), whereas sharp creases are observed in experiments before that limit.

The problem is difficult because it involves self-contacting singular solutions of non-linear (finite strain) elasticity equations, and has resisted many theoretical attempts.

The new analytical prediction agrees well with experiments in the literature, and with numerical results. It is clear that it brings an important answer to a longstanding question.

The article is clearly written, avoiding technical details, rather focusing on the general approach towards the result.

One question that was not addressed is the assumption of neo-hookean material, which is one specific type of material response. How general are the results presented here?

It would be interesting to explain how the analytical method applies to other types of elastic law, and how the threshold are modified. Were experiments performed with perfectly neo-hookean materials?

I support publication of the paper.

Author's reply to the remarks on the manuscript NCOMMS-17-28515-T

This letter contains the point-by-point reply to the remarks of the reviewers.

The extracts of the reviewer's letter are reported using a normal font, whilst the author's reply is written using an italic font.

Reviewer #1 (Remarks to the Author):

This paper address the theory of creasing in highly compressed elastic solids. Creasing occurs when a soft solid body with a free surface is compressed. At a critical degree of compression, the free surface transitions from being flat to having "sharp self-contacting folds ... known as creases". Creasing is ubiquitous in the mechanics of soft solids, and of considerable importance. However, although creasing has been well understood numerically via finite element calculations, there is no analytic theory of creasing. This is surprising, since it appears to be part of the family of elastic instabilities, and there is a standard theoretical toolbox for understanding elastic instabilities based on perturbative linear stability analysis, but these tools fail to capture creasing. The difficulty is easy to identify: a centre of a crease is a cusp with divergent strain/stress, which is not a small perturbation from the flat state of the free surface, even if the amplitude/displacement of the crease is very small. Creasing is thus a new type of "non-linear instability", and finding an analytic description of creasing is certainly one of the most important open questions in the theory of mechanical instabilities, both because creasing itself is important, and because it appears to be a new category of instability.

I am sincerely grateful to the reviewer for his/her careful consideration of this work, the thoughtful review and the constructive criticism. In the following, I will respond to the technical issues on a point-by-point basis.

Here the author claims to have found an analytic description of crease nucleation. The centerpiece of this theory is a prediction of the degree of compression required to nucleate a crease which, for the simplest (neo-Hookean) constitutive law the author is able to produce a surd (14) which amounts to compression by 36.2..%. This prediction is convincingly close to the values that previous studies have found numerically. However, although I can follow the calculation presented by the author to produce this value, I don't understand how the calculation connects to the nucleation of creases.

To be more precise, the author imagines the center/cusp of a crease as following a well known solution in elasticity that corresponds to taking a semi-circle of (2-D) elastic material, and bringing together the two radii to form the semi-circle into a circle. Similarly, far from the crease, the author asserts the elastic body has a homogeneous compression, which is the "outer" solution. For small degrees of compression, the entire body is homogeneously compressed, but when the threshold for creasing is reached, a crease nucleates consisting of an central region of the "inner solution" which somehow matches onto the outer homogeneous solution. This picture was introduced in the second paper by Hohlfeld and Mahadevan, and is likely correct.

To derive the threshold for cusping, the standard approach would be to consider the linear stability of the homogeneous solution, since this is the solution occupied by the elastic body prior to creasing, which becomes unstable at the creasing threshold. This was done by Biot in the 1960s, who showed that the homogeneous solution becomes catastrophically unstable at about 45% compression, however the creasing occurs earlier, at around 35% compression, when the homogeneous solution is certainly linearly stable. Here the author instead considers the stability of

the "inner" solution. He considers a semicircle of material wrapped into a circle of radius r_c , following the well-known inner solution, with the point at r_c being the point where the crease ceases self contact and starts to open up. This solution has an unknown overall offset in its pressure field, which the author determines by demanding that the stress at the point r_c match the macroscopic compressive stress in the homogeneous solution. The author then adds a small displacement perturbation that looks like $\cos(m\theta)$, where $0 < \theta < 2\pi$ is the angular coordinate in the circular state. The author linearizes the equations of elastic equilibrium within the inner solution, and finds the form of small perturbation that solves the equations to linear order. He then considers the "boundary" of the inner solution as a circle, and applies the boundary condition that the stress on the boundary after perturbation should be the same as the stress in the unperturbed inner-solution state but at the perturbed location of the boundary.

The author solves the above stability problem, producing the curve Fig 1 in the SI which maps the compressive stress in the homogeneous solution required to make the inner solution unstable for each value of m .

The reviewer rightfully suggests that the article would should include a clearer explanation on how the proposed calculation connects to the crease nucleation.

I revised the text accordingly. In particular, I added a paragraph explaining the peculiar characteristics of a 'material instability' such as creasing, for the benefit of a general audience journal not necessarily expert in instability theories.

I will discuss the more technical details in the response to the following points.

I have reproduced this calculation, and it is correct, though I believe there are a few serious typos in the SI.

SI eqn 9/10 -> this equation is a linearization in U , but has a $4UU'$ term in?

SI eqn 11 -> I think this is wrong by an overall minus sign?

SI eqn 12 -> I think there are several errors here, and consequently the surd in (13) doesn't solve (12) for $m=2$? I think 12 should be " $m(20(-a+b)+29(-a+b)m^2+(-6+7a+7b)m\sqrt{40+9m^2})-4(10(a-b)m+7(-a+b)m^3-(1+a)(1+b)\sqrt{40+9m^2}+(1+a+b)m^2\sqrt{40+9m^2})-8(a-b)m(-1+m^2)\sqrt{40+9m^2}=0$ ", which does indeed give the authors surd as a root for $m=2$.

I am very thankful to the reviewer for checking all the steps in the theoretical derivation and for spotting these typos, that have now been corrected. I also found that the expression in eq. 11 and 12 within SI could be further simplified.

Despite these typos, the calculation does seem to be correct. However, fundamentally I don't understand what the solution to this stability problem has to do with the creasing instability.

The thesis of this work is that creasing is a material instability, which means that it is local in space and does not depend on the shape of the body. At the same time, it is mathematically associated with the loss of strong local minimum, being a global instability in the configurational space. Creasing is therefore different from buckling, which is associated with the loss of weak local minimum and it is therefore a structural instability, i.e. shape dependent, local in the configurational space while being global in the actual space. A structural instability can be detected by considering perturbations that are small together with their derivatives. In the post-buckling regime, such unstable modes can localize in space and have higher local energy than the trivial state (such as in B. Audoly's Phys. Rev. E 2011, and H. Diamant, T.A. Witten Phys. Rev. Lett. 2011). However, the global

energy will be necessarily lower. In the creasing problem, the trivial solution is instead stable with respect to weak variations, however, it becomes unstable with respect to local perturbations with small amplitudes but non-small derivatives. The energy of such perturbations is instead smaller than in the trivial solution both locally and globally in space.

In particular, I would ask the author to address the following questions:

1) Why does the stability of the inner solution relate to the nucleation of creases at all? The inner solution is only seen after the crease has formed, so what does its stability have to do with anything?

It is shown that the zero-order inner solution can never match the outer solution, since a circle can never match the elliptic cavity resulting from the perturbed outer solution. Thus, the threshold for the loss of the radial symmetry of the inner solution represents a necessary condition for crease nucleation. It is also sufficient if the perturbed inner solution (that has now an elliptic shape) can match the far away displacement and pressure fields of the outer solution. Thus, by showing that such matching is possible we satisfied necessary and sufficient conditions for the existence of a creased solution.

Can you make a clear claim that the threshold derived here is an upper bound, lower bound or actual calculation of the threshold for the crease forming instability?

It is difficult to give a sharp answer to this question. The trivial upper and lower bound are now extensively discussed in order to guide the readers to the metastable regime of interest for creasing and to the meaning of stability with respect to weak and strong variations. The derived threshold is introduced as a refined upper bound for the creasing instability, because the inner and outer solutions cannot be matched for higher levels of stretch. Notwithstanding, it can also be considered as a lower bound because the theoretical construction provides the actual destabilizing perturbation at this predicted value of compression. Thus, in my understanding it can be regarded as an actual calculation of the threshold. However, before the actual mathematical theorem is proved using formal arguments of calculus of variation, nothing can be claimed for sure. Indeed, there may be other, completely different destabilizing perturbations (i.e. test functions in different functional spaces) which would lower the value of the critical threshold, and in this case this threshold will be only an upper bound.

However, the comparison with numerical simulations shows that such a predicted threshold would be remarkably good.

The \sim sign in (13) leaves me unsure how precise a claim is being made.

I agree that the \sim sign could be misleading, and it has been replaced by \approx , meaning that I have a precise surd prediction that is almost equal to 0.637554. This is now discussed in the revised text to avoid confusion.

I also don't understand the analogy to a heteroclinic trajectory in a dynamical system - what is the actual argument for the stability of the inner solution being relevant?

The creased solution locally reduces the strain energy in the nucleation zone but cannot match geometrically the outer solution. Thus, there must exist an inhomogeneous solution connecting these two inner and outer states in an intermediate domain. Here I vaguely resort to the analogy with heteroclinic orbits connecting two fixed points in dynamical systems for illustrative purposes. The fixed points represent two solutions of the original problems none of which agrees with all side conditions. A typical example is a traveling wave solution describing a moving domain boundary in the Ginsburg-Landau equation or a reaction front in the Fisher-Kolmogorov equation. This analogy is used here to make a bridge with these classical results of dynamical systems and it is not involved in any technical steps. The existence of such an inhomogeneous “heteroclinic trajectory” is shown in the article by construction of the matching solution linking the perturbative series representing the inner and outer solutions.

All the considerations are now discussed in the revised text, trying to establish the best possible balance between technical argumentations and ease of reading for a general audience public. I also added a reference to an experimental work of Tang et al. appeared in Soft Matter in 2017, where they refined the measurements of the crease nucleation threshold using high speed cameras and a fluorescent monomer, providing a critical stretch value of 0.635, that differs only by 0.0027 with respect to the proposed analytic threshold.

2) Where do the boundary conditions on the inner solution come from? Why are they the appropriate boundary conditions for the stability analysis, rather than, say, that the boundary stress does not change upon perturbation, or that the stress match the intermediate solution? Similarly, why is a circular (/semicircular) boundary the appropriate boundary to consider?

The reviewer raises a fundamental point that deserves further discussion. The choice of the circular (resp. semicircular) boundary in the current (resp. reference) configuration is dictated by the assumption that the inner solution is given by Eq. 7. This guess is compatible with the symmetry of the problem and with the experimental observation of a self-contacting surface (not simply a cusped profile as in free boundary fluid flows). More interestingly, Eq.7 belongs to a class of universal solution, which means that is a solution that can be found independently on the constitutive laws of the material under consideration by applications of surface tractions alone. This explain why creasing is ubiquitous in soft materials.

The choice of the boundary condition for the stability problem of the inner solution means exactly that the boundary stress does not change after perturbation. In this sense it is a dead load and not a pressure load. It comes from a simple Taylor expansion on the normal stress considering that the boundary is now perturbed:

$$\sigma_{rr}(r_0 + \epsilon U_r \cos(m\theta)) = \sigma_{rr}(r_0) + \sigma'_{rr}(r_0) \cdot \epsilon U_r \cos(m\theta) = \sigma_{out} + \delta\sigma$$

Thus, the overall equilibrium of the slab is globally preserved after perturbation, since the far out stress applied on the side walls remains unchanged.

3) In the manuscript, you write "The basic inner solution is given by (7), whilst the axial symmetry and global equilibrium impose $\sigma_{rr}(r_c, \theta) = \sigma_{out}$." I can't see why global equilibrium imposes this - can you elaborate? Indeed, I would have thought $\sigma_{\theta\theta}(r_c, \pi)=0$ would be more appropriate, since this is the point the self-contact ends and the surface becomes a free surface?

For the sake of simplicity, imagine to cut the slab along the vertical axis passing through the self-contacting line. The global equilibrium of the half-slab imposes that the overall horizontal force along the midline must counterbalance the far-out horizontal stress σ_{out} , without creating any overall angular moment.

Let r_0 be the radius of the incipient crease, the overall horizontal force exchanged along the midline of the crease is given by:

$$2 \int_0^{r_0} \sigma_{\theta\theta} dr = 2 \int_0^{r_0} (\sigma_{rr} r)_{,r} dr = 2\sigma_{rr}(r_0) r_0$$

If $\sigma_{rr}(r_0) = \sigma_{out}$, then this zero order inner solution and the zero-order outer solution along the midline perfectly equilibrate the far field horizontal stress. The boundary conditions on the incremental inner and outer problems also ensure that this global equilibrium is preserved. In this sense, not only the terms of the perturbative series respect the local equilibrium conditions at each order of approximation, but the overall matched solution proposed here also fulfils the global equilibrium.

Nonetheless, imposing $\sigma_{\theta\theta}(r_0, \pi)=0$ would be another rightful option, as assumed in Hertzian contact. I had earlier considered it myself, but then I could not derive mechanically consistent results. Let me sketch the basic reasoning behind this incongruence. From Eq. 7, the radial and hoop stresses at (r_0, π) are given by:

$$\begin{aligned} \sigma_{\theta\theta}(r_0) &= \mu\lambda_\theta^2 - p(r_0) = 2\mu - p(r_0); \\ \sigma_{rr}(r_0) &= \mu\lambda_r^2 - p(r_0) = \mu/2 - p(r_0) = \sigma_{\theta\theta}(r_0) - 3/2\mu. \end{aligned}$$

Now if we assume that $\sigma_{\theta\theta}(r_0, \pi)=0$, then:

$$\sigma_{rr}(r_0) = -3/2\mu < \sigma_{out}(\lambda_{cr}) \simeq -2.05\mu$$

This means that the overall horizontal stress exchanged along the crease midline is much lower than the horizontal stress that is imposed on the lateral walls at the critical stretch threshold.

Thus, in order to enforce the global equilibrium in the horizontal direction, an extra-stress must be exchanged along the slab midline. But this is not physically achievable, since the incremental stress δS_{xx} of the outer solution along the midline (i.e at $x=0$) is always zero (it can be checked by substituting eqs. 17-18 of the article and eq. 15 of the SI in the definition of incremental stress δS_{xx} in eq. 9). Thus, this option must be discarded.

The numerical simulations apparently confirm that the proposed boundary condition is correct. Looking at the Figure 4 (left) of SI, the pressure field at the crease radius is found at about

2.463μ (this can be checked with the colorbar at the cusp point). This value is very close to the one expected close to the nucleation threshold, since:

$$p(r_0) = -\sigma_{rr}(r_0) + \mu\lambda_r^2 = -\sigma_{out} + \mu\lambda_r^2 \simeq (2.05 + 1/2)\mu = 2.55\mu$$

4) Why do you not consider an $m=1$ mode, which becomes unstable at $\sigma_{out}=-\sqrt{7}/2=-1.32\dots$ (i.e. before the $m=2$ mode) or indeed an $m=1/2$ mode which might have the effect of opening out the inner solution into a crease? I understand that $m=1$ breaks reflection symmetry, but surely the symmetry should emerge from the calculation not be fed into it?

It can be shown that the mode $m=1$ is always stable, so that the symmetry emerges as a result of the calculations. Looking at the reviewer's prediction $\sigma_{out}=-\sqrt{7}/2=-1.32$, I recognized the signature of a mistake I also did in a preliminary stage of this work.

In fact, the value of c_1 given by eq. (11) of SI would diverge for $m=1$, since the denominator goes to zero. It can indeed be shown that the correct value is $c_1=0$, and that that the resulting dispersion relation never vanishes. Thus, the critical mode is $m=2$.

Moreover, even if non integer wavenumbers would correspond to a crease opening, these situations are physically not relevant. First, they would correspond to a cusped solution without a crease. As discussed in the previous points, the emergence of a creased nucleus of finite size is indeed essential to reduce the overall strain energy, as confirmed by experiments. Second, such a crease opening is out of reach for this perturbative analysis. Taking $m=1/2$, for example, one would obtain an unphysical result: a vanishing incremental hoop stress superposed to a finite hoop stress along the opening line, that should instead be free of normal tractions.

The result that the mode $m=2$ is the critical one seems to be confirmed by numerical simulations (e.g. please refer to the figure 2 of Hohlfeld's PRL in 2013; where the inset with a x40000 zoom clearly shows the existence of an elliptic creased domain).

In the latter part of the manuscript, the author discusses matching the inner solution to the outer solution by finding an intermediate solution to the equations of elasticity, in a manner reminiscent of the resolution to the Stokes paradox. Although this is surely a good idea, I'm skeptical about the implementation here. Firstly, the intermediate solution is taken as the Green's function for a point force applied at the middle of the crease. The stress fields in this Green's function transmit an actual, nonzero, vertical force through the elastic body, as can be seen by integrating the stress around the surface of any portion of the body including the force, yet there is no actual vertical force applied at the heart of a crease, so any attempt to match this intermediate solution to the inner solution will end up having stress divergence (i.e. force) at the interface between the solutions - I would have thought a force dipole would be a better matching solution? The author doesn't run into this problem because, as far as I can see, he never actually matches the inner and intermediate solutions since, as seen in (20), the range of validity of the intermediate solution ($r > r_c$) does not overlap with the range of validity of the inner solution ($r \sim r_c$).

I am grateful to the reviewer for acknowledging the novelty of the proposed approach, and I agree that the matching deserves further discussion.

The core point is the following: when the stretch reaches the critical threshold for crease nucleation, the elastic boundary problem changes radically, since the free surface is no longer traction-free because of self-contact. Since self-contact is frictionless, there is an exchange of a surface traction

that is normal to the self-contacting line. The exchanged traction gives zero net force in the current configuration, but if one pulls it back to the homogeneously deformed state, a vertical distributed pressure is obtained on a flat portion of the free surface in that homogeneously deformed configuration. At a certain distance from the creased area, this vertical traction can be approximated by a small concentrated force F superposed on the homogenous horizontal stress σ_{out} . Thus, the resulting vertical force in the perturbed outer solution balances the traction exchanged by self-contact after the finite rotation giving rise to the inner creased solution.

Let me now discuss in further detail the range of validity of the matching. The inner and intermediate solution cannot be matched directly at r_c , especially since I approximated a distributed self-contacting traction with a concentrated force. Using the proposed approximation, there is perfect analogy with the Stokes problem: the intermediate solution perceives the crease as a point, and the matching with the inner solution can only be performed by introducing an intermediate variable in a given range of validity away from the self-contacting domain.

Overall, I have very mixed thoughts about this paper. On one hand, the calculation clearly draws together relevant components of the problem, and the match between the predicted and numerically measured threshold is convincing. On the other hand, I simply don't understand how the author's stability analysis relates to the problem of crease nucleation. Beyond this general lack of understanding, I'm particularly confused about how/why the boundary conditions are chosen, how the inner solution pressure constant is chosen, and how the intermediate solution can ever match the inner or outer solution given it implies a body force within the crease which isn't actually there.

This is a sufficiently important problem that, despite my doubts about the correctness of this work, I am very sympathetic to the idea it should be published for the community to reflect on, and Nature Communications is a reasonable journal for it to be published in. However, in my view the author needs to clarify the above issues and make a clear physical argument for why this calculation relates to crease nucleation. Alternatively, the author could simply address the more technical questions (2-3-4 above, and the question about the body force implied by the intermediate solution) and then add some prominent caveats to the manuscript, walking back a little from the claim that "This work solves the quest for a theoretical explanation of crease nucleation" and present it as a potentially important step towards a theoretical solution, but one that remains fundamentally mysterious.

I am very grateful to the reviewer for the careful consideration of this work, the constructive criticisms and the words of appreciation. I certainly agree that this analysis is not definitive and some points, such as the determination of the crease depth and the surface profile of the crease opening, remains unanswered. Accordingly, I have some sentences in the revised text to state the limitations of this approach in a clear and fair manner. On one hand, I revised the manuscript in order to make the technical points as clear as possible without introducing technical jargon. On the other hand, I rephrased some claims in order to highlight what is achieved here and what needs to be achieved in future efforts.

Reviewer #2 (Remarks to the Author):

This article reports analytical study of the formation of creases at the surface of an elastic soft solid under compression. The article also includes comparison with of the results with a numerical simulation. The author provides here an analytical prediction for the threshold of creasing. This is interesting because the system was shown to be linearly stable up to a large compression (Biot, 1963), whereas sharp creases are observed in experiments before that limit. The problem is difficult because it involves self-contacting singular solutions of non-linear (finite strain) elasticity equations, and has resisted many theoretical attempts.

The new analytical prediction agrees well with experiments in the literature, and with numerical results. It is clear that it brings an important answer to a longstanding question.

The article is clearly written, avoiding technical details, rather focusing on the general approach towards the result.

I support publication of the paper.

I sincerely thank the reviewer for the positive comments on this work.

One question that was not addressed is the assumption of neo-hookean material, which is one specific type of material response. how general are the results presented here?

It would be interesting to explain how the analytical method applies to other types of elastic law, and how the threshold are modified. Were experiments performed with perfectly neo-hookean materials?

The reviewer raises an interesting point that has been further discussed in the revised text and SI. Most experiments in literature (included the one depicted in Figure 1) are performed on soft gels, that behave as neo-Hookean materials (since the elastic network is made by long monomers cross-linked in water, so that entropic elasticity dominates). However, few experimental works use rubber materials (such as the works of Gent on crease nucleation whilst bending a thick block). Although this article calculates the nucleation threshold for a neo-Hookean body, the proposed criterion for creasing is applicable to every constitutive equation for an isotropic, hyperelastic solid. In fact, the inner deformation is a universal solution in finite elasticity, meaning that it can be supported in equilibrium for every isotropic material by suitable surface tractions alone. Thus, the creasing threshold specific to a different constitutive equation can be calculated by solving the corresponding inner incremental problem, exactly as it is shown in this work.

Reviewers' comments:

Reviewer #1 (Remarks to the Author):

I would like to thank the author for reacting constructively to my previous report. The author has satisfied me on most of the issues I raised, and I am now happy for the paper to proceed to publication.

I am still very unsure that the results in the paper are correct, but I think, at this stage, it is best for them to be published so the wider community can consider them. However, I will take this final opportunity to put to the author two points that I still find particularly troubling:

1) The author has some sense of "global equilibrium" between the inner and outer solution, that is never spelt out, and which I still don't understand. This sense enters crucially both in setting the constant in the pressure field for the inner solution, and in setting the boundary condition for the perturbation of the inner solution.

The author makes some attempt to spell it out in his reply to my first report, explaining that the constant in the stress is set so that the force transmitted through the vertical line bisecting the crease in the inner solution (total length $2r_0$) must be the same as that applied to the body in a vertical length $2r_0$ at the slab's boundary. I just don't see why this should be true - which piece of boundary is the crease matching to? The top $2r_0$? The $2r_0$ that segment at the same vertical level? The total length of bisecting line is less than the total length of the boundary. Why not match to the segment of the boundary that was at the same horizontal level in material coordinates rather than final coordinates?

Furthermore, the author's argument that the choice of constant in the inner pressure agrees with the numerical images in the SI does not match my interpretation of the same images - I think the crease appears to cease self contact around the boundary of the darkest and second darkest blue, suggesting a pressure of around 1.687μ , or much closer to the value of 1.5μ needed to set $\sigma_{\theta\theta}=0$ than the value claimed in the text. Furthermore, the author's argument seems to require that $\sigma_{\theta\theta}$ is discontinuous at the end of the self-contact, since it goes from finite at r_c to zero just above r_c , but there is no sign of such a discontinuity in the numerics.

2) In the matching solution, the author invokes a downwards force acting at the center of the crease. Taking the Cauchy (final state) stress as σ , the equation of equilibrium is

$$\text{div}(\sigma)=f_{\text{body}}$$

where the divergence is taken in final state coordinates and the forces are real, final state forces. If one integrates σ_{rr} from the matching solution around a (final-state) semi-circle from free surface to free surface far from the crease but fully including it, one learns, from the divergence theorem, that there is a final state downwards force acting on the material within crease. However, in reality there is no such force, so the "matching" solution will never match the inner solution.

I queried this in my first report, and the author replied that this force is the contact force in the crease "pulled-back" to the reference/material configuration, where it would indeed be a vertical force. However, I don't think this explanation adds up: the Cauchy stress argument above deals with final state forces and areas, with no need to pull back anything, and remains valid, so the problem remains. In contrast, if we consider the "pulled-back" PK2 stress (which is entirely in material or "pulled-back"

coordinates) the equation of equilibrium is

$$\text{div}(\mathbf{F}, \mathbf{PK2}) = \mathbf{f}_{\text{body}}$$

where the divergence is done in material coordinates, but \mathbf{f}_{body} is evaluated in the final state. Thus applying a divergence theorem argument to the same contour but with PK2 gives the same answer - that the matching solution is associated with a vertical force at the crease in the final state. The "pulled-back" force does not feature in the argument.

Reviewer #2 (Remarks to the Author):

My previous report was in favor of publication. The author has addressed my suggestion concerning the generalization to other non-linear laws. I have however an additional comment.

After reading the new version and the answers to referee n.1, I appreciate better the nature of the analytical argument for the condition of existence of a creased solution. It is a key point in the article. Not being a specialist in stability analysis, I had not realized that the argument is very non-standard. The loss of marginal stability for the creased inner solution is presented as an indication that another solution exist that bridge the inner and outer solutions.

I feel that the author could explain this step still further:

For example, in the analogy with the heteroclinic solution of ODE, the two fixed point play a similar role. Following this line of thought, the existence of the heteroclinic solution would therefore lead to a loss of marginal stability for both the inner and the outer solutions at the same time. This is however not so (the outer solution is only unstable at higher compression of the Biot threshold). I am obviously missing a point.

I therefore believe that the status of this stability analysis as a proof of existence should be explained further. I don't mean necessarily mathematically justified, but rather put into the general context of mathematical stability analysis.

I do recommend publication as the threshold obtained through this argument is relatively simple, is analytical, and compares convincingly with numerics and experiments in this difficult highly non-linear problem.

Author's reply to the remarks on the manuscript NCOMMS-17-28515A

This letter contains the point-by-point reply to the remarks of the reviewers.

The extracts of the reviewer's letter are reported using a normal font, whilst the author's reply is written using an italic font.

Reviewer #1 (Remarks to the Author):

I would like to thank the author for reacting constructively to my previous report. The author has satisfied me on most of the issues I raised, and I am now happy for the paper to proceed to publication.

I am grateful to the reviewer for the continued interest in this work and its positive evaluation.

I am still very unsure that the results in the paper are correct, but I think, at this stage, it is best for them to be published so the wider community can consider them. However, I will take this final opportunity to put to the author two points that I still find particularly troubling:

1) The author has some sense of "global equilibrium" between the inner and outer solution, that is never spelt out, and which I still don't understand. This sense enters crucially both in setting the constant in the pressure field for the inner solution, and in setting the boundary condition for the perturbation of the inner solution.

The author makes some attempt to spell it out in his reply to my first report, explaining that the constant in the stress is set so that the force transmitted through the vertical line bisecting the crease in the inner solution (total length $2 r_0$) must be the same as that applied to the body in a vertical length $2 r_0$ at the slab's boundary. I just don't see why this should be true - which piece of boundary is the crease matching to? The top $2 r_0$? The $2 r_0$ that segment at the same vertical level? The total length of bisecting line is less than the total length of the boundary. Why not match to the segment of the boundary that was at the same horizontal level in material coordinates rather than final coordinates?

Before nucleating a crease, the slab is subjected to a homogeneous uniaxial stress of magnitude σ_{out} in the horizontal direction, that is maintained through the uniform traction exerted at the side walls. The assumption of this work is that the average stress per unit length transmitted through the vertical line bisecting the crease for both the inner and outer solutions is also equal to σ_{out} .

I acknowledge that the justification referring to a "global equilibrium" may be confusing and should better clarified. The manuscript clearly states that the final length of the bisecting line is out of reach of this perturbative approach unless higher order terms are considered. In the revised text it is now highlighted that this assumption ensures that the average horizontal stress exchanged in each portion of the bisecting line pertaining to the inner and outer solutions is equal to the far-out stress. This ensures that in each domain of validity of the perturbative series there is a global balance of forces and linear momentum.

Furthermore, the author's argument that the choice of constant in the inner pressure agrees with the numerical images in the SI does not match my interpretation of the same images - I think the crease appears to cease self contact around the boundary of the darkest and second darkest blue, suggesting a pressure of around 1.687μ , or much closer to the value of 1.5μ needed to set $\sigma_{\theta\theta}=0$ than the value claimed in the text. Furthermore, the author's argument seems to require that $\sigma_{\theta\theta}$ is discontinuous at the end of the self-contact, since it goes from finite at r_c to zero just above r_c , but there is no sign of such a discontinuity in the numerics.

This work does not contain any claim about a discontinuity in the hoop stress. In order to avoid any further misunderstanding, let me clarify that the crease opening is not assumed to occur exactly at r_c . The revised text states that the release of hoop stress in proximity of the crease opening is out of reach of this model, since it occurs in a domain situated outside the ranges of validity of both the inner and the intermediate solution. By looking at the same numerical image in the SI, we can certainly agree that the crease opening creates a boundary layer effect on the pressure (and thus the required vanishing of the hoop stress) over a characteristic distance that is much smaller than the crease radius.

Let me add a further supporting argument. In my previous reply, I underlined that by assuming $\sigma_{\theta\theta}(r_c) = 0$ the overall horizontal stress exchanged along the vertical crease midline would be much lower than the horizontal stress that is imposed on the same portion of the lateral walls at the critical stretch threshold. This would imply the occurrence of an extra horizontal stress exchanged across the vertical bisecting line far away from the inner creased domain, that is not observed in simulations. Moreover, if the guess of $\sigma_{\theta\theta}=0$ was correct, the corresponding value of pressure should be found not only at the surface opening point but also all around the nucleation point at about the same distance from the crease centre. Again, this is contradicted by the numerical results.

2) In the matching solution, the author invokes a downwards force acting at the center of the crease. Taking the Cauchy (final state) stress as σ , the equation of equilibrium is $\text{div}(\sigma)=f_{\text{body}}$

where the divergence is taken in final state coordinates and the forces are real, final state forces. If one integrates σ_{rr} from the matching solution around a (final-state) semi-circle from free surface to free surface far from the crease but fully including it, one learns, from the divergence theorem, that there is a final state downwards force acting on the material within crease. However, in reality there is no such force, so the "matching" solution will never match the inner solution. I queried this in my first report, and the author replied that this force is the contact force in the crease "pulled-back" to the reference/material configuration, where it would indeed be a vertical force. However, I don't think this explanation adds up: the Cauchy stress argument above deals with final state forces and areas, with no need to pull back anything, and remains valid, so the problem remains. In contrast, if we consider the "pulled-back" PK2 stress (which is entirely in material or "pulled-back" coordinates) the equation of equilibrium is $\text{div}(F.PK2)=f_{\text{body}}$

where the divergence is done in material coordinates, but f_{body} is evaluated in the final state. Thus applying a divergence theorem argument to the same contour but with PK2 gives the same

answer - that the matching solution is associated with a vertical force at the crease in the final state. The "pulled-back" force does not feature in the argument.

I agree with the reviewer that the proposed perturbative analysis deserves further discussion. The presence of both a stress singularity at the crease center and of a discontinuity of the deformation tensor at the free surface greatly complicates the mechanical interpretation.

Here I proposed a non-standard procedure in order to build a perturbative series of the outer solution taking into account for the traction exerted across the self-contacting line. Since the inner deformation is known, this traction is pulled back and superposed to the homogeneously deformed configuration mapped by zero-th order outer solution (as shown in Fig 3 B in the SI). As rightfully pointed out by the reviewer, the resulting matching solution around a (final-state) semi-circle from free surface to free surface far from the crease but fully including it is associated with a net vertical force. However, this force just balances the mechanical effects provoked by the self-contact at the free surface, represented by the concentrated force.

In simple terms: a localised finite rotation of the free surface creates a self-contact, that transforms the uniaxial state of horizontal compression of the outer solution into a localised creased state characterized by a biaxial state of stress. By pulling back the traction exerted along the self-contacting free surface, the perturbative outer series takes into account for the vertical compression due to the localised surface folding superposed over the homogeneous state of horizontal compression. The upward net vertical force in the matching solution indeed balances the downward concentrated force exerted by the localised folding of the free surface.

In summary, this perturbative approach simply represents the mechanical state of the creased solution at a reasonable distance from the crease nucleation point. A direct matching is simply not possible since the first-order incremental displacement fields become of the same order of the corresponding zero-order terms when r approaches r_c , thus the solution no longer represents a meaningful perturbative series.

I am sincerely thankful to the reviewer for the thought-provoking questions and the genuine interest in this problem.

Reviewer #2 (Remarks to the Author):

My previous report was in favor of publication. The author has addressed my suggestion concerning the generalization to other non-linear laws. I have however an additional comment.

After reading the new version and the answers to referee n.1, i appreciate better the nature of the analytical argument for the condition of existence of a creased solution. It is a key point in the article. Not being a specialist in stability analysis, i had not realized that the argument is very non-standard. The loss of marginal stability for the creased inner solution is presented as an indication that another solution exist that bridge the inner and outer solutions. I feel that the author could explain this step still further: For example, in the analogy with the heteroclinic solution of ODE, the two fixed point play a similar role. Following this line of thought, the existence of the heteroclinic solution would therefore lead to a loss of marginal stability for both the inner and the outer solutions at the same time. This is however not so (the outer solution is only unstable at higher compression of the Biot threshold). I am obviously missing a point. I therefore believe that the status of this stability analysis as a proof of existence should be explained further. i don't mean necessarily mathematically justified, but rather put into the general context of mathematical stability analysis.

A heteroclinic trajectory in ODEs is an orbit of the phase space such that it escapes one fixed point and enters into another fixed point. Thus, a necessary condition for its existence is that at least one fixed point is unstable in a dynamical sense, i.e. has an eigenvalue with non-negative real part. For example, an inhomogeneous traveling wave solution of the Fisher Kolmogorov equation exists because one fixed point is a stable node/spiral and the other is a saddle. This draws the parallel to the creasing problem: since the outer solution is always marginally stable before the Biot threshold, then it is necessary that the inner solution be marginally unstable.

However, I preferred not to add further explanations of this analogy in the revised text for avoiding confusion to the reader, possibly arising for the difference in the notion of stability for dynamical systems and for an elastic quasi-static problem. The most fundamental point is that creasing involves the loss of a strong local minimum, thus it is a material shape-independent instability intrinsically different from a structural shape-dependent instability, such as buckling.

I do recommend publication as the threshold obtained through this argument is relatively simple, is analytical, and compares convincingly with numerics and experiments in this difficult highly non-linear problem

I am sincerely grateful for the careful consideration of this work and the positive comments.